

# The University of Washington Ice-Liquid Discriminator (UWILD) improves single particle phase classifications of hydrometeors within Southern Ocean clouds using machine learning

Rachel Atlas[1], Johannes Mohrmann[1], Joseph Finlon[1], Jeremy Lu[1,*], Ian Hsiao[1,*], Robert Wood[1], and Minghui Diao[2]

[1]Department of Atmospheric Sciences, University of Washington
[2]Department of Meteorology and Climate Science, San Jose State University
[*]These authors contributed equally to this work

**Correspondence:** Rachel Atlas (ratlas@uw.edu)

**Abstract.** Mixed-phase Southern Ocean clouds are challenging to simulate and their representation in climate models is an important control on climate sensitivity. In particular, the amount of supercooled liquid and frozen mass that they contain in the present climate is a predictor of their planetary feedback in a warming climate. The recent Southern Ocean Clouds, Radiation and Aerosol Transport Experimental Study (SOCRATES) vastly increased the amount of in-situ data available from
mixed-phase Southern Ocean clouds useful for model evaluation. Bulk measurements distinguishing liquid and ice water content are not available from SOCRATES so single particle phase classifications from the Two-Dimensional Stereo (2D-S) probe are invaluable for quantifying mixed-phase cloud properties. Motivated by the presence of large biases in existing phase discrimination algorithms, we develop a novel technique for single particle phase classification of binary 2D-S images using a random forest algorithm, which we refer to as the University of Washington Ice-Liquid Discriminator (UWILD). UWILD uses
14 parameters computed from binary image data, as well as particle inter-arrival time, to predict phase. We use liquid-only and ice-dominated time periods within the SOCRATES dataset as training and testing data. This novel approach to model training avoids major pitfalls associated with using manually labelled data, including reduced model generalizability and high labor costs. We find that UWILD is well calibrated and has an overall accuracy of 95% compared to 72% and 78% for two existing phase classification algorithms that we compare it with. UWILD improves classifications of small ice crystals and large
liquid drops in particular and has more flexibility than the other algorithms to identify both liquid-dominated and ice-dominated regions within the SOCRATES dataset. UWILD misclassifies a small percentage of large liquid drops as ice. Such misclassified particles are typically associated with model confidence below 75% and can easily be filtered out of the dataset. UWILD phase classifications show that particles with area-equivalent diameter ($D_{eq}$) < 0.17 mm are mostly liquid at all temperatures sampled, down to -40°C. Larger particles ($D_{eq}$ > 0.17 mm) are predominantly frozen at all temperatures below 0°C. Between 0°C and
5°C, there are roughly equal numbers of frozen and liquid mid-size particles (0.17 < $D_{eq}$ < 0.33 mm) and larger particles ($D_{eq}$ > 0.33 mm) are mostly frozen. We also use UWILD's phase classifications to estimate sub 1-Hz phase heterogeneity and we show examples of meter-scale cloud phase heterogeneity in the SOCRATES dataset.



# 1 Introduction

## 1.1 Southern Ocean cloud phase and climate

Mixed-phase processes within Southern Ocean clouds moderate cloud radiative effects (Bodas-Salcedo et al., 2016; McCoy et al., 2014a) and cloud-climate feedbacks associated with the stormy region (McCoy et al., 2014b). The presence of small amounts of ice within liquid-dominated mixed-phase clouds can substantially increase precipitation as compared with warm clouds with similar thickness, due to efficient cold precipitation formation (Bergeron, 1928; Field and Heymsfield, 2015). Increased precipitation can reduce cloud lifetime (Albrecht, 1989) and increase aerosol scavenging (Radke et al., 1980).

The distribution of liquid and frozen hydrometeors within Southern Ocean clouds will change as the climate warms (Mitchell et al., 1989; Storelvmo et al., 2015). Climate models that simulate a relatively high ice to liquid ratio within Southern Ocean clouds in the present climate, or base state, exhibit a negative cloud radiative feedback in future climates (McCoy et al., 2015). One explanation for this is that as the climate warms, fewer ice particles form, and clouds become both brighter (Sun and Shine, 1994) and longer lived. This feedback, known as the cloud-phase feedback, has a large impact on the climate sensitivity and its weakening has been suggested as an explanation for an increase in climate sensitivity going from CMIP5 to CMIP6 (Zelinka et al., 2020; Bjordal et al., 2020).

Given that the strength of the cloud-phase feedback in climate models is related to the base state representation of cloud phase within Southern Ocean clouds, we can constrain the cloud-phase feedback by assessing how realistic the base state of each climate model is. Historically, due to a lack of in-situ measurements over the Southern Ocean, satellite data have been used to evaluate cloud phase within climate models. Tan et al. (2016) retrieved cloud phase from Cloud-Aerosol Lidar with Orthogonal Polarization (CALIOP) to show that climate models with strong cloud-phase feedbacks typically underestimate the fraction of supercooled liquid in present-day extra-tropical, mixed-phase clouds. McCoy et al. (2014a) combined retrievals from the International Satellite Cloud Climatology Project (ISCCP), Moderate Resolution Imaging Spectroradiometer (MODIS), and Multiangle Imaging Spectroradiometer (MISR) to connect seasonally varying cloud radiative fluxes with mixed-phase cloud properties. Satellite products have been invaluable for constraining cloud radiative effects and identifying model biases over the Southern Ocean but they provide almost no information on hydrometeor phase other than at the cloud top. For this reason, and because of low vertical and horizontal resolution, and large retrieval uncertainties, satellite products are not sufficient to support process-oriented studies of mixed-phase microphysics in models.

Recent in-situ measurements of summertime Southern Ocean clouds from the Southern Ocean Clouds, Radiation, Aerosol Transport Experimental Study (SOCRATES; McFarquhar et al., 2020) make it possible to quantify cloud microphysical properties through the full depth of the cloud and to create a dataset for evaluating models and remote sensing retrievals. Since measurements of bulk ice water content (IWC) are not available during SOCRATES, single particle classification is the most viable way to quantify ice properties within SOCRATES-sampled clouds. Furthermore, single particle phase classifications are useful for making direct comparisons of simulated and observed liquid and frozen particle size distributions (PSDs). During





SOCRATES, optical array probes (OAPs) such as the Two-Dimensional Stereo (2D-S) instrument (Lawson et al., 2006) collected binary images of single particles that can be used for phase classification. However, there is no standard procedure for identifying the phase of 2D-S imaged particles, and existing algorithms, which are described in the following subsection, have substantial biases. Here, we introduce a novel machine-learning based algorithm, the University of Washington Ice-Liquid

Discriminator (UWILD), and show that it has greater skill in discriminating liquid and ice particles than two pre-existing algorithms.

### 1.2 Existing methods for phase classification

Single particle phase classification techniques for binary OAP images typically distinguish liquid from frozen hydrometeors using particle shape and/or particle roughness. Cober et al. (2001a) used four different ratios computed from particle length,

width, perimeter and area to estimate particle sphericity and discriminate between liquid and ice particles. McFarquhar et al. (2013) proposed using area ratio, defined as the projected area divided by the area of the smallest circle circumscribing the particle, to classify the phase of particles with maximum dimensions between 35 and 60 $\mu$m. Using area ratio alone is the simplest technique for phase classification and has been implemented in the Earth Observing Laboratory's OAP processing code. Throughout this study, we compare UWILD's performance with this technique, which we refer to simply as "Area

Ratio". We include a schematic of Area Ratio in Figure 1a.

A limitation of using area ratio alone is that quasi-spherical frozen particles, such as graupel, may be classified as liquid. Using particle surface roughness is an attractive alternative. Czys and Schoen Petersen (1992) fit a fourth degree polynomial to quasi-spherical 2D particle images (with particle circumference represented in polar coordinates) to estimate surface roughness and distinguish between graupel particles and liquid drops. Other studies (Hunter et al., 1984; Moss and Johnson, 1994; Bower

et al., 1996; Yang et al., 2016) have used Fourier analysis to quantify the sphericity and roughness of imaged particles and to classify them as liquid drops or frozen particles with particular habits. Holroyd (1987) also used a combination of particle shape and surface roughness to classify particle habit. They quantified surface roughness using the fine detail ratio, defined as the perimeter multiplied by maximum dimension and divided by projected area, and used that and other particle properties to classify frozen hydrometeors into ten habits, which we refer to as Holroyd habits. McFarquhar et al. (2018) modified this

algorithm for use in mixed-phase clouds by classifying particles labeled as tiny or spherical from the Holroyd (1987) scheme as liquid and all other particles as ice. Furthermore, for particles with maximum dimensions $< 300$ $\mu$m, the presence or absence of a light spot in the center of the particle, known as the Poisson spot (Arago and Gay-Lussac, 1819; Heymsfield and Parrish, 1978), is recorded. If particles exhibit this diffraction pattern, they are classified as liquid and their Holroyd habit is not taken into account. This technique has been applied to SOCRATES data (Wu et al., 2019) and Wang et al. (2020) used the

resulting phase classifications to compute number concentrations and water contents of droplets and frozen hydrometeors in SOCRATES-sampled clouds. Throughout this study, we compare UWILD's performance with this technique as well, which we refer to as "Holroyd" for simplicity although it employs the Holroyd (1987) habit classification in conjunction with other techniques. We include a schematic of Holroyd in Figure 1b.



Machine learning is attractive for single particle phase classification because it allows for the use of multiple particle parameters without incurring the labor and time costs associated with hand picking thresholds for those parameters. Additionally, many machine learning algorithms produce both a classification and a model confidence for each particle, which can be regarded as the uncertainty in the phase classification for a well-calibrated model. Machine learning has been used to classify airborne particle probe images in many studies over the last decade (Lindqvist et al., 2012; Nurzynska et al., 2012, 2013; Praz et al., 2017, 2018; Xiao et al., 2019; Wu et al., 2020; Korolev et al., 2020; O'Shea et al., 2016; Touloupas et al., 2020), but to our knowledge, it has never been used to directly predict hydrometeor phase for binary OAP images.

The remainder of the article is arranged as follows. In section 2, we describe the SOCRATES dataset and the development of our training, testing and validation dataset. In section 3, we introduce UWILD and in section 4, we evaluate UWILD and compare its performance with two pre-existing phase classification algorithms. In section 5, we show how concentrations of liquid and frozen hydrometeors vary with size and temperature, and investigate sub 1-Hz cloud phase heterogeneity in the SOCRATES dataset. A summary of UWILD and its implications for identifying particle phase in Southern Ocean clouds is discussed in section 6.

## 2 Processing SOCRATES data for single particle phase classification

### 2.1 SOCRATES Observations and description of the 2D-S

Between 15 January and 25 February 2018, the U.S. National Science Foundation (NSF) supported the SOCRATES campaign to sample diverse boundary layer clouds between Hobart, Tasmania and the Antarctic continent (McFarquhar et al., 2020). SOCRATES employed the NSF/National Center for Atmospheric Research (NCAR) Gulfstream V (G-V) aircraft outfitted with four particle imaging probes including the 2D-S, the Two-Dimensional Cloud (2DC) probe, the Particle Habit Imaging and Polar Scattering (PHIPS) probe and the Precipitation Imaging Probe (PIP). The SOCRATES campaign included fifteen 6-8 hour flights. We use 14 out of the 15 SOCRATES flights, omitting RF15 due to an anomalously high occurrence of corrupted 2D-S particle images. Figure 2 shows the SOCRATES flight tracks and the distribution of 1-Hz in-cloud temperatures from the 14 SOCRATES flights used here. 85.5% of the cloudy flight data occur within the temperature range that can support mixed-phased clouds, -40° to 0°C, while 15.1% of the cloudy flight data are warmer than 0°C, and 0.4% of the flight data are colder than -40°C.

The PHIPS records high quality images of particles, with a maximum imaging rate set at 3 Hz for SOCRATES, and measures particle scattering phase functions at a maximum rate of 3.5 kHz (Abdelmonem et al., 2016; Schnaiter et al., 2018). Both particle images and phase scattering functions from the PHIPS can be used for phase classification but only the particle images have particle size information (Waitz et al., 2020). The lower imaging rate of the PHIPS is not suitable for developing 1-Hz distributions of particle properties so the PHIPS was not considered for this study.

The 2D-S, 2DC and PIP are OAPs which return shadow images of particles. The 2DC and the PIP both suffered from quality control issues (Finlon et al., 2020) so we use the 2D-S particle images to develop single particle phase classifications for the SOCRATES dataset. The 2D-S uses a 128-element photodiode array in conjunction with high-speed electronics to generate





shadowgraphs of particles with 10-$\mu$m optical resolution as they enter the instrument's sample volume (Lawson et al., 2006). Particle shadowgraphs, or images, are composed of 10-$\mu$m x 10-$\mu$m square pixels. Particles with maximum dimensions smaller than 0.01 mm have areas smaller than that of a single pixel and cannot be detected by the 2D-S. Particles with maximum dimensions greater than 1.28 mm (the length of the photodiode array) have a higher likelihood of being partially cut-off by the image buffer depending on their habit and orientation. The 2D-S has a sample volume between 10 and 16 L s$^{-1}$ for typical SOCRATES aircraft speeds and can record compressed data at rates associated with particle concentrations up to about 100 cm$^{-3}$ (Lawson et al., 2006), as they typically were during SOCRATES. We use the University of Illinois/Oklahoma OAP Processing Software (McFarquhar et al., 2018) to compute morphological properties of individual particles (e.g., maximum dimension, area equivalent diameter, perimeter, projected area, area ratio, and habit) from the horizontal channel of the 2D-S. Due to optical limitations of the 2D-S, quasi-spherical out-of-focus liquid particles exhibiting Poisson spots are size corrected following Korolev (2007). We apply several criteria to filter particles for use in this study. First, we only use particles whose center is within the 2D-S field of view to minimize uncertainties in determining the reconstructed particle size (Heymsfield and Baumgardner, 1985; Field, 1999). Due to uncertainties in defining the probe's depth of field and sample area (Baumgardner and Korolev, 1997; Jackson et al., 2012; Heymsfield and Parrish, 1978) and limited shape information to support robust classification for smaller particles (Korolev et al., 1991; Baumgardner et al., 2017), particles with fewer than 25 pixels are excluded from this study. Because of this, there are no particles classified as tiny (defined as < 25 pixels) using the Holroyd scheme (Figure 1a) and only the right subtree of Area Ratio (Figure 1b) is relevant. Throughout this study, we will use area-equivalent diameter ($D_{eq}$) to represent particle size. For the 2D-S, $D_{eq}$ in mm is computed from the number of pixels as follows:

$$D_{eq} = 2 \times 10^{-2} \times \sqrt{\text{number of pixels} \times \pi^{-1}}$$

We use 15 morphological features, listed in Table 1, to train our machine learning model and we show histograms of these features for the SOCRATES dataset in the appendix. We use the common logarithm of inter-arrival time (time elapsed since the previous particle was imaged within the 2D-S sample volume) instead of the absolute inter-arrival time because machine learning models are optimized to train on normally distributed variables. Inter-arrival time is the only variable that is substantially non-normally distributed and requires normalization. In the SOCRATES dataset there is generally a distinct mode of particles with very short inter-arrival times ($10^{-6}$ to $10^{-7}$ seconds) likely due to shattered ice particles (not shown). This mode is not present in the histogram of log(inter-arrival time) shown in Figure A1 because the tiny shattered ice particles have been removed by our minimum size filter. Thus, we do not apply additional filters to remove shattered particles from the dataset.

## 2.2 Preparation of training, validation and test data

Classification problems are a type of supervised machine learning that require a dataset with known classifications to use for model training, cross-validation during hyperparameter (i.e. model configuration parameter) tuning, and model testing. Creating this dataset, hereafter referred to as the TTV (for train-test-validate) set, is the biggest challenge associated with this particular machine learning problem. Using manual inspection to build a TTV set, as most studies using machine learning for





particle image classification have, would limit the scope of our TTV set to particles large enough to identify by eye that have
an unambiguous phase. This would result in our TTV set being substantially different from a set of randomly sampled particles

from SOCRATES, and reduce the generalizability of our machine learning model for the whole SOCRATES dataset. Instead,
we use in-situ flight data, including temperature from the HARCO heated total air temperature sensor (EOL, 2019), water
vapor mixing ratio from the Vertical Cavity Surface Emitting Laser (VCSEL) hygrometer (Zondlo et al., 2010; Diao, 2020)
and voltage from from the Rosemount Icing Detector (RICE) (EOL, 2019), to identify flight periods where the hydrometeors
are most likely to be all or mostly the same phase. Equations of saturation pressure with respect to liquid and ice (Murphy and

Koop, 2005) were used to calculate relative humidity with respect to liquid (RH) and ice ($RH_i$), respectively. Uncertainties of
RH and $RH_i$ can be derived based on the uncertainties associated with temperature and water vapor. Uncertainties range from
6.4% to 6.8% for RH, and from 6.5% to 6.9% for $RH_i$ from 0°C to -40°C, respectively. Identifying liquid phase regions of cloud
is simple because frozen hydrometeors rarely persist at temperatures above 5°C (Yuter et al., 2006; Oraltay and Hallett, 2005).
Thus, we select a five minute flight period where the temperature varies between 6°C and 12°C as a liquid-only period. We

show time series of 1-Hz temperature, RH, particle count, and liquid fraction from UWILD for this region in Figure 3a. There
is only RH data available towards the end of the period and the RH is close to 100% there. UWILD classifies most particles as
liquid throughout the flight period, but its accuracy decreases towards the end of the period. We explain how UWILD classifies
particles in Section 3 and we quantify UWILD's performance and identify biases in its classifications in Section 4. A histogram
of temperature for the liquid-only period is shown in Figure 2b and a normalized histogram of area-equivalent diameter ($D_{eq}$)

is shown in Figure 4. Small particles ($< 100$ pixels or $D_{eq} < 0.1$ mm) dominate the liquid-only dataset.

Supercooled water can persist at all temperatures above the homogeneous nucleation threshold of -40°C (Korolev et al.,
2017) and below 0°C. Since there are no multi-second periods of in-cloud data from SOCRATES with temperatures below
-40°C (Figure 2b), we cannot be certain that all particles within any given SOCRATES flight period are frozen. However, we
can use the aforementioned atmospheric parameters and particle probe images to identify periods where we have very high

confidence that over 99% of the particles are frozen. We refer to these as ice-dominated periods. We use temperature, RH with
respect to ice and liquid water, voltage from the RICE, and particle images from the PHIPS and the 2D-S to identify two ice-
dominated periods, which we show in Figures 3b and 3c. The ice-dominated periods are defined as having no RICE response,
are sub-saturated with respect to liquid, and supersaturated with respect to ice. A RICE response, which consists of the 1-Hz
voltage oscillating over a 20-second period, is expected when the supercooled liquid water content exceeds 0.01 g m$^{-3}$ (Biter

et al., 1987). RF04 is the source of 70% of particles in the combined ice-dominated dataset and RF01 is the source of 30%. A
histogram of temperature for the combined ice-dominated dataset is shown in Figure 2b and a normalized histogram of $D_{eq}$ for
the same dataset is shown in Figure 4. The ice-dominated dataset is composed primarily of medium-size and large particles ($\geq$
100 pixels or $D_{eq} \geq 0.1$ mm) particles.

We manually inspected 500 2D-S particle images from the combined ice-dominated dataset with $D_{eq} \geq 0.2$ mm and found

that all of them are unambiguously frozen. We also manually inspected 500 randomly sampled images with $D_{eq} < 0.2$ mm,
which account for 13% of the particles in the ice-dominated dataset, and found that up to 1% of the particles are unambiguously
liquid and an additional 1% could be liquid but are unidentifiable by eye. Thus, as many as 0.3% of the particles in the ice-



dominated regions may be liquid. If UWILD classified all particles in the ice-dominated region correctly, it would have a slightly higher performance than what is reported here (Section 4) because we compute performance metrics assuming that all
particles in the ice-dominated region are frozen.

We also use phase classifications from the PHIPS to evaluate our liquid-only and ice-dominated periods. The PHIPS dataset includes both automated classifications using the scattering phase function and manual classifications using particle images. Because of the greater maximum scattering data acquisition rate compared to the maximum imaging rate described in the beginning of this section, there are more automated classifications than manual classifications. The PHIPS automated classification
algorithm identified 132 particles as liquid and 45 particles as frozen during our liquid-only period, but manual classifications of 106 images are universally liquid. This bias in the PHIPS algorithm arises from the fact that large liquid drops are typically aspherical because they are distorted due to pressure differences in the instrument's inlet and are thus misclassified as ice (Fritz Waitz, personal communication). While the PHIPS was not available for RF01, the PHIPS algorithm automatically classified 3905 particles as frozen and only 4 particles as liquid during our ice-dominated period from RF04. Manual classifications of
320 images are universally frozen.

We compare the liquid-only period and two ice-dominated periods with two examples of mixed-phase periods in Figure 3. The first mixed-phase period (Figure 3d) samples a stratocumulus cloud within the boundary layer. The RICE voltage oscillates throughout the period, indicating the presence of supercooled water exceeding 0.01 g m$^{-3}$, and the liquid fraction is greater than 75% most of the time. The cloud is saturated with respect to liquid. The second mixed-phase period (Figure 3e) sampled
near the top of altostratus cloud. This period is colder and, on average, sub-saturated with respect to liquid water and saturated with respect to ice. The RH is sub-saturated even though the cloud is liquid-dominated in the sampled region because the aircraft is skirting a horizontally variable cloud top. The liquid fraction is close to 1.0 and the RICE voltage oscillates until 00:23:00 UTC ($\sim$ 0.4 UTC), when the liquid fraction decreases abruptly and the RICE voltage stabilizes. The change in phase occurs because the aircraft is initially sampling the cloud top and transitions to sampling below the cloud top.

The liquid-only period includes 90,000 particles that pass our size threshold, while the ice-dominated periods include 55,000 particles. All particles drawn from the liquid-only period are labelled liquid and all particles drawn from the ice-dominated regions are labelled ice. These labels are taken as truth for the purposes of model training and evaluation (Section 4). We partition the particles into three size categories: small (corresponding to 25-99 pixels or D$_{eq}$ of approximately 0.056-0.1 mm), medium (100-699 pixels or 0.1-0.3 mm), and large ($>$ 700 pixels or $>$ 0.3 mm). In the remainder of this study, all references to
particle size will be in terms of D$_{eq}$. We then randomly subsample the liquid particles down to have an equal total number of ice and liquid particles, preserving the ratio between the 3 size categories. These particles are then partitioned into training (60%), test (20%), and validation (20%) sets, again preserving the original ratios between the three size bins for each phase separately, as well as the balance of particles from each phase. We refer to the combined training, test, and validation sets as the TTV set. We explicitly preserve these rough size distributions to ensure that the test set has a reasonable number of small ice crystals and
large liquid drops for evaluation, as these particles are rare enough in the full TTV dataset that a completely random partition risks having them undersampled in the test set. Preserving a balance between liquid and ice particles simplifies interpretation of model performance summary statistics; this is further discussed at the beginning of Section 4. The composition of the TTV set





is broken down in Table 2. We show histograms of particle properties from the TTV set and in the whole SOCRATES dataset (14 flights) in the appendix, and discuss out of sample particles.

## 3   UWILD: Description and interpretation

A key consideration for all machine learning applications is the choice of machine learning model. One approach to analyzing particle probe images is to apply deep learning directly to the captured image (e.g. Touloupas et al., 2020; Xiao et al., 2019; Wu et al., 2020; Korolev et al., 2020). Here, we take a simpler approach and employ a random forest model (Breiman, 2001; Pedregosa et al., 2011), which requires a preprocessing step to extract relevant image features (e.g., particle area or max diameter; full list in Table 2). Classification is then carried out using these features as inputs (Lindqvist et al., 2012; Nurzynska et al., 2012, 2013; Praz et al., 2018, 2017; O'Shea et al., 2016). An advantage of this approach is that it allows for the inclusion of features not directly related to particle appearance; in particular, we show that inter-arrival time is a valuable discriminator of liquid and ice particles. This variable would be more difficult to incorporate in an image-based deep learning model. Random forests can also provide more interpretable results, as the trained model can be analyzed to investigate relative feature importance. Another advantage (shared by many machine learning approaches) is the determination of classification confidence, which can be useful in filtering particles depending on application, or estimating uncertainties in calculated bulk properties such as liquid water content.

For a decision tree trained using a supervised learning approach, the training set is split by thresholding features (e.g. whether particle area ratio is more or less than 0.8); precisely which feature and which value is determined by whatever 'best' splits the dataset into distinct categories (for UWILD the max Gini impurity reduction criterion is used). This process is repeated on each data subset until the data are entirely partitioned into distinct categories. In a random forest, multiple such trees (100 for UWILD) are trained using random subsets of data features. Randomness is introduced here to reduce overfitting to the training set and improve model generalizability. For a given test datapoint, each tree provides a classification, and the plurality vote of all trees is the overall category assigned to the datapoint, with the proportion of trees voting for that category as the model confidence. We include a simple schematic of UWILD in Figure 1c.

A model is well-calibrated if its model confidence (internal prediction probability) accurately reflects its performance. Figure 5 shows this relationship between model confidence (from the random forest votes) and model accuracy (how likely the model was to correctly classify particles), evaluated on the test set. A one-to-one relationship is ideal because it indicates that we can directly use model confidence as an estimate of prediction uncertainty. For example, a particle classified as ice with a model confidence of 75% should be seen as 75% likely to be ice, and 25% likely to be liquid. Figure 5 also shows that UWILD has a confidence of 95% or higher for 73% of the particles in the TTV set.

To better understand how the UWILD classifier determines particle phase, we quantify how much it relies on each of the fifteen different features (listed in Table 1) using permutation feature importance analysis. This technique measures how much a model relies on the information encoded in a particular feature by calculating model accuracy on a test set, randomly shuffling a given feature, and measuring how much the accuracy decreases. The random shuffling of a feature renders that feature useless



to the model classification and the model accuracy will decrease substantially for a very significant feature. This analysis can be rapidly performed multiple times for each feature. Another advantage of permutation feature importance is that it is a function of the dataset being used for evaluation as well as the model, and so this metric can be calculated separately for different subsets of the data, or for entirely new test datasets. Other measures of feature importance (such as impurity-based feature importance)

are functions only of the model and do not share this advantage. A relevant drawback to all measures of feature importance is that they are affected by correlations between features. As correlated features share information, model performance may not decrease as much when a particular feature is shuffled, as a (previously) correlated different feature may still encode the relevant information. However, decorrelating variables prior to use, which would address this issue, complicates model interpretation (while not significantly affecting model performance), and so we chose to preserve original features, and caution against too

minute a dissection of the permutation feature importance.

Figure 6 shows the permutation feature importance of the top ten features, split by particle size and evaluated on the model test set. Overall, we see that width, area ratio, and log(inter-arrival time) are the most important. The next two features (max dimension and length) both encode size and correlate well with width, while the remaining features have low impact on model accuracy. Put another way, the model primarily relies on these first three features for classification. Considering differences

between size classes, we note that width (and in fact all size-related features) are most important for medium particles, which is to be expected as larger particles are predominantly ice and smaller particles predominantly liquid, with medium particles varying the most. For small particles, log(inter-arrival) time is most important, and for large particles, area ratio is most important (likely because small and medium particles are mostly quasi-spherical irrespective of phase). Regarding correlated features, the results in Figure 6 should not be taken to mean that particle width in particular is a key discriminator as opposed to length

or max dimension, but rather that the width is a good estimator of particle size, which is the particle characteristic that matters in determining its phase. If particle width were removed from the feature set, then another size-encoding feature would appear more important.

## 4   Comparison between phase classification schemes

For quantitative evaluation of a classification model, an intuitive summary metric is model accuracy (the ratio of correct

classifications to total classifications). The overall accuracy of UWILD, Holroyd, and Area Ratio on our test set is 94.9%, 78.5% and 71.8%, respectively, indicating that UWILD is performing quite well. However, accuracy is most suitable for balanced classification problems (i.e., when data are spread evenly across categories). In the case of highly unbalanced problems, high accuracy can be achieved by systematically erring in favor of the dominant category. For example, small particles in the test set are overwhelmingly liquid (Figure 4) so high accuracy can be achieved by predicting that all small particles are liquid at

the expense of correctly classifying small ice particles.

Model performance, especially for unbalanced classification problems, can be better measured by calculating precision (the ratio of all particles correctly classified as liquid to all particles classified as liquid) and recall (the ratio of all particles classified as liquid to all true liquid particles). Both scores range from 0-1, and they penalize false positives and false negatives,





respectively, for a particular category. These scores are unified in the F1 score, which is their harmonic mean:

$$F1 = \frac{2 \times \text{precision} \times \text{recall}}{\text{precision} + \text{recall}}$$

The F1 score is a conservative measure of model performance because the lesser of recall and precision will dominate the harmonic mean, and it can be calculated for various data subsets. We show the F1 scores for Holroyd, Area ratio and UWILD in Figure 7 as a function of phase and size class. This analysis is performed on our test set. UWILD outperforms Holroyd and Area Ratio for all phases and size classes. It has the best performance for small liquid (F1=0.982) and large ice (F1=0.992)

particles and performs less well with small ice (F1=0.765) and large liquid (F1=0.893) particles. While UWILD performs least well when classifying small ice and large liquid, it nevertheless has a particularly large performance advantage over Holroyd and Area Ratio for those categories. Holroyd outperforms Area Ratio for medium and large ice particles, whereas Area Ratio outperforms Holroyd for small ice particles and liquid particles of all sizes. In the rest of this section, we identify differences between the algorithms and biases within each algorithm to explain these discrepancies in their performances. We use the

whole SOCRATES dataset (14 flights), which includes 5.76 million classified 2D-S images, for our analysis for the rest of the paper.

Table 3 shows how many particles each algorithm classified as liquid and ice for each size class, and in total. In general, Holroyd classifies the most particles as ice and Area Ratio classifies the most particles as liquid. UWILD and Area Ratio both classify over 90% of the small particles as liquid whereas Holroyd classifies only 70% of them as liquid. Area Ratio classifies

three quarters of the medium particles and half of the large particles as liquid. In contrast, the other two algorithms classify about 40% of the medium particles and 0% (Holroyd) to 2.7% (UWILD) of the large particles as liquid.

In Figure 8, we show the fraction of particles classified as liquid, from the three phase discrimination algorithms, in the phase spaces of temperature vs particle size (left column) and RH vs particle size (right column). In the second row, we show a 2D histogram of the confidence from UWILD, and in the third, fourth and fifth rows, we show 2D histograms of

the fraction of particles classified as liquid by the three phase discrimination algorithms. At temperatures greater than -20°C, UWILD confidence is lowest in areas where UWILD transitions between having a high liquid fraction and a low liquid fraction (Figure 8b). UWILD confidence is also low for small particles at temperatures below -20°C which can have high or low liquid fractions. All three algorithms show a decrease in liquid fraction for small particles at temperatures between -20°C and -30°C and an increase in the liquid fraction at temperatures below -30°C (Figure 8c-e). This behavior is a consequence of small

sample size, as the liquid dominated data below -30°C comes from just one flight that sampled the top of an altostratus cloud, whereas the ice-dominated data at higher temperatures come from several flights that sampled the middle of altostratus clouds.

Since temperature and RH are not inputs to any of the algorithms, we can use them to gauge whether the particle classifications make physical sense. In other words, we can use these atmospheric parameters to make broad predictions of hydrometeor phase and determine which algorithm is most consistent with these predictions. We expect that small particles will be entirely

liquid above 0°C and that large particles will be primarily liquid above 0°C and entirely liquid above 5°C due to having longer melting timescales (Oraltay and Hallett, 2005).





Ice and liquid precipitation formation mechanisms have been observed to operate simultaneously at temperatures as low as -28°C (Huffman and Norman, 1988; Cober et al., 2001b; Kajikawa et al., 2000; Korolev et al.; Silber et al., 2019) so we cannot use temperature alone to make a prediction for the liquid fraction of particles at temperatures below 0°C. However, we do

expect to see a size dependence in the liquid fraction. To our knowledge, the largest liquid particle associated with supercooled drizzle formation (as opposed to melting frozen hydrometeors) that has been noted in the literature has a maximum dimension of 0.625 mm (Cober et al., 2001b). Most SOCRATES data were collected in conditions that could not support the re-lofting of melted frozen hydrometeors. Furthermore, melted frozen hydrometeors are rarely lofted to temperatures below -5°C in environments that do support re-lofting (Oraltay and Hallett, 2005). Thus, we expect that medium-sized and large droplets at

temperatures below -5°C are primarily formed via supercooled drizzle formation and will not be present at the largest sizes (0.625 - 1 mm).

Holroyd and UWILD classify too many medium-sized (0.1 mm < $D_{eq}$ < 0.3 mm) and large ($D_{eq}$ > 0.3 mm) particles as ice at warm temperatures and the bias is more pronounced for Holroyd. Liquid fractions for Holroyd sharply decrease to near 0.0 for particles with $D_{eq}$ between 0.2 and 0.3 mm at all temperatures (Figure 8e). This strong size dependence arises from the fact

that Holroyd only considers the presence or absence of a Poisson spot for particles with maximum dimensions less than 0.3 mm (Figure 1b). Particles exceeding that maximum dimension threshold must be nearly spherical in shape to be classified as liquid because the presence of a Poisson spot is not factored into the phase classification. UWILD's liquid fraction for particles with $D_{eq}$ between 0.2 and 0.5 mm at temperatures between 0°C and 5°C is approximately 0.5 (Figure 8c). This is unrealistically low particularly for the warmer end of this temperature range, where most frozen particles would have melted.

Holroyd also classifies many small particles ($D_{eq}$ < 0.1 mm) as ice (Figure 8e) at all temperatures. Its liquid fraction never exceeds 0.86 for small particles at temperatures above -20°C, whereas Area Ratio and UWILD have a liquid fraction near 1.0 (Figure 8c,d). Holroyd's relatively low liquid fractions for small particles are unrealistic for temperatures above 0°C.

Area Ratio classifies too many large particles ($D_{eq}$ > 0.3 mm) as liquid at cold temperatures (Figure 8d). Area Ratio's liquid fraction rarely drops below 0.8 for particles with $D_{eq}$ > 0.2 mm and temperatures below -5°C, whereas Holroyd and UWILD

have liquid fractions near 0.0 (Figure 8c,e). While a liquid fraction between 0.5 and 0.8 for these particles is not physically impossible, the fact that there is no decrease in the liquid fraction with increasing particle size for particles with $D_{eq}$ between 0.5 mm and 1 mm, where particle sizes and temperatures are inconsistent with supercooled drizzle formation, implies that Area Ratio's higher liquid fractions may be unrealistic.

There are also clear differences between the three algorithms' classifications in RH vs particle size space (Figure 8h-j).

UWILD and Area Ratio both have liquid fractions near 1.0 for small particles near liquid saturation (RH = 100%), whereas Holroyd has a liquid fraction closer to 0.75 for the same region. Uncertainty in RH is around 7% so while high liquid fractions are most common at liquid saturation they occur at a wide range of RH values. Additionally, fluctuations in RH from dry air entrainment and in-cloud circulation can lead to deviations from liquid saturation at 1-Hz resolution. UWILD classifies fewer particles as liquid in subsaturated air than either Area Ratio or Holroyd. In the midsize particle range (0.1 < $D_{eqc}$ < 0.2

mm), which includes drizzle, the liquid fraction is near 1.0 when the RH is close to 100% and it drops down to 0.2 when RH decreases to 50%.





Liquid particles can persist in subsaturated air at or below the cloud base, and these regions were purposefully sampled within the boundary layer during SOCRATES. Drizzle drops falling below liquid clouds evaporate in the subsaturated environment, reducing their size. Subsaturated air can also be associated with ice-dominated clouds as $RH_i$ is higher than RH throughout the
mixed-phase temperature range. In ice-dominated clouds, cloud droplets are produced at the turbulent cloud top and tend to freeze before forming drizzle drops. For both of these reasons, we expect a decrease in the average size of liquid particles as the RH decreases below liquid saturation. UWILD is the only algorithm for which the 50% liquid fraction shifts to smaller sizes as the RH falls below 100%. Thus, UWILD's lower liquid fractions in regions with RH < 100%, for particles in the midsize particle range $(0.1 < D_{eq} < 0.2$ mm), are more realistic than Holroyd's and Area Ratio's higher liquid fractions.

UWILD is the only algorithm of the three that can achieve liquid fractions near 0.0 and near 1.0, in both temperature vs particle size space (Figure 8c) and RH vs particle size space (Figure 8h). Thus, it has the flexibility to represent both the liquid-only regions that we expect at the warmest temperatures and near liquid saturation, and the ice-dominated regions that we expect at the coldest temperatures and the largest particles sizes, and in subsaturated regions.

The dashed boxes labelled A-D on the 2D histograms in the left column of Figure 8 highlight areas of disagreement between
the models, whereas box E highlights agreement between the models regarding the presence of supercooled water at -35°C. Figure 9 shows randomly sampled images from each of the five regions within the dashed boxes. Each particle image has the UWILD confidence printed above the particle and the phase classifications from all three algorithms printed below the particle. Since we have chosen to primarily focus on areas of disagreement between the algorithms, there are more misclassifications in these regions than in the dataset as a whole.

Box A highlights a region where Area Ratio has a liquid fraction near 1.0 across all size categories, Holroyd has a liquid fraction of about 0.75 for small particles and 0.0 for large particles, and UWILD has a liquid fraction of 1.0 for the warmest temperatures and 0.5 for temperatures near 0°C. Since temperature ranges from 0°C to 10°C here, we expect that the particles are mainly liquid although large ice particles can persist warmer than 0°C. Furthermore, quasi-spherical frozen particles can have a close resemblance to large liquid drops and the two cannot necessary be distinguished by eye from 2D-S images.

Randomly sampled particles from this region appear to be predominantly liquid due to the absence of rough edges along the perimeter or non-spherical habits (Figure 9a). Out of 50 randomly sampled images, Area Ratio classifies one particle as ice, Holroyd classifies 21 particles as ice, and UWILD classifies 12 particles as ice. Of the sampled particles classified by UWILD with a confidence $\geq$ 75%, all but one of them appear to be properly identified as liquid. The only exception is one particle with a confidence of 81%, which is likely misclassified due to the particle being truncated at the end of the image buffer and
thus yielding a lower area ratio. A greater proportion of particles with a confidence below 75% are classified as ice by UWILD but are likely liquid, comprising about 55% of the sampled particles for these lower confidences. The misclassifications can be removed from UWILD, if desired, by filtering out particles that have a confidence of less than 75% and/or are touching the edge of the image buffer. Holroyd misclassifies 9 more liquid particles as ice than UWILD, but does not provide a measure of confidence that can be used to assess the likelihood of mis-classification. Of the particles that UWILD likely misclassifies as
ice, many have particularly large Poisson spots. Particle area is computed from shadowed diodes exclusively so Poisson spots





are not included. Additionally, particles with Poisson spots are resized following Korolev (2007). Both of these factors affect the calculation of area ratio and, thus, the phase classification of the particle.

Box B highlights a temperature and size range where UWILD and Area Ratio are in agreement that the liquid fraction is near 1.0 but Holroyd has a lower liquid fraction of about 0.75. The 50 randomly sampled images shown appear to be mostly

370 liquid with some irregular small ice crystals also present. UWILD performs better for these temperatures and particle sizes than in Box A; it classifies 4 particles as ice, of which 1 is clearly liquid and 3 are unidentifiable by eye. Holroyd classifies 20 particles as ice, of which most are likely misclassifications.

Box C and D both highlight regions where UWILD and Holroyd are in agreement that the liquid fraction is near 0.0 but Area Ratio has a higher liquid fraction. The discrepancy is larger for box D, where Area Ratio has a liquid fraction of about 0.75.

375 Out of 45 randomly sampled images, Area Ratio misclassifies eight quasi-spherical ice crystals as liquid for Box C and at least 13 quasi-spherical ice crystals as liquid for box D. Particles in box C primarily come from boundary layer clouds and most have columnar habits with smaller area ratios although there are a smaller number of large quasi-spherical frozen particles with large area ratios present as well. Many of the particles in box D come from the atmospheric river case described in Finlon et al. (2020) and, thus, are more likely to have quasi-spherical heavily rimed habits with larger area ratios. The different particle

380 habits explain the discrepancy in Area Ratio's performance in the two different regions.

In both boxes C and D, UWILD and Holroyd likely misclassify several large particles as ice and those particles are largely associated with low confidence in UWILD. It is difficult to quantify this bias because there are several particles that could be either quasi-spherical frozen particles or large droplets and cannot be distinguished by eye. This does not mean that UWILD cannot classify those particles because it uses inter-arrival time in addition to image-derived parameters to make classifications.

385 Box E highlights a region where all three phase discrimination algorithms have liquid fractions greater than 0.5 despite sampling very cold temperatures (-33°C to -36°C). Randomly sampled images with high confidence in UWILD have spherical habits and most have Poisson spots as well, suggesting that the particles in this region are primarily liquid. UWILD and Area Ratio classify all 20 randomly sampled images as liquid whereas Holroyd classifies 4 particles in the sample as ice. These particles are particularly small and lack Poisson spots so we cannot identify their phase by eye. Nevertheless, it is clear that all

390 three algorithms correctly identify this region as liquid-dominated. These particles were sampled during a period from RF03 that is plotted in the second half of Figure 3e, where the aircraft skirted the top of an altostratus layer.

## 5 Applications

### 5.1 Particle size distributions

We use UWILD's classifications and confidences to compute median 1-Hz liquid and frozen particle size distributions (PSDs)

395 and uncertainties for all SOCRATES data (14 flights). We show average PSDs for five different temperature ranges, and for the whole dataset, in Figure 10. The x-axis is $D_{eq}$ (consistent with other figures) and the y-axis is the particle concentration normalized by the log of the bin width. Both axes are plotted on a log scale. The dashed lines are deterministic distributions which means they are generated using the UWILD classifications without taking the model confidence into account. All clas-





sified particles are used for this analysis regardless of model confidence. The solid lines and shaded areas around them are the

median and interquartile range of 30 bootstrapped samples which are generated using the model confidence. For example, if a particle is classified as ice with 75% confidence than it is considered an ice particle for the deterministic distribution but, on average, it will be considered an ice (liquid) particle in 75% (25%) of the bootstrapped samples. Note that due to the log scale on the y-axis, the effect of bootstrapping is mainly noticeable where concentrations are small, and is strongest where the differences between ice and liquid concentrations span orders of magnitude. Note also that the bootstrapped distributions

fall entirely between the deterministic distributions. This can be understood by considering, for example, the smallest particles just below 0°C, where there are approximately 100 times as many liquid particles as ice. If, when taking into consideration model confidence, 2% of the liquid particles are reclassified as ice, this is barely noticeable in the liquid particle concentration, but results in a tripling of the ice particle concentration (Figure 10c). As the bootstrapped PSDs are better representations of the true PSDs, considering model confidence is most essential in the areas in Figure 10 where there are large discrepancies

between the deterministic and bootstrapped distributions (e.g. estimating sub-mm ice particle concentrations around 0°C).

Within the SOCRATES dataset, small particles ($D_{eq} < 0.1$ mm) are more likely to be liquid at all temperature ranges but they have much higher concentrations and are more liquid-dominated above -20°C (Figure 10c,d,e). The concentrations of medium-sized (0.1 mm $< D_{eq} < 0.3$ mm) and large ($D_{eq} > 0.3$ mm) particles decrease as temperature increases. Medium-sized particles are ice-dominated between -40°C and -20°C (Figure 10a) and liquid-dominated above -5°C (Figure 10d,e). Large

particles are liquid-dominated above 5°C (Figure 10e), where drizzle formation becomes the dominant mode of precipitation. The largest particles ($D_{eq} > 1$ mm) are ice-dominated at all temperatures but have small concentrations ($< 2 \times 10^{-4}$ cm$^{-3}$) above 5°C (Figure 10e). The cross-over point, or the $D_{eq}$ at which the PSDs transition from being liquid-dominated to being ice-dominated, is 0.1 mm for temperatures between -40°C and -20°C (Figure 10a), 0.17 - 0.33 mm for temperatures between -20°C and 5°C (Figure 10b,c,d), and 0.7 mm at higher temperatures (Figure 10e). The cross-over point for the whole dataset is

0.17 mm (Figure 10f), which is in agreement with the phase classifications from the PHIPS for all SOCRATES flights (Waitz et al., 2020, figure 8). Small ice crystals and large liquid drops are associated with the most uncertainty at all temperature ranges.

In Section 4, we showed that UWILD misclassifies some large liquid particles as ice. We examined 200 randomly sampled 2D-S images of particles with $D_{eq} > 0.16$ mm from temperatures between 0°C and 5°C, and found that 16% of particles clas-

sified as ice are actually liquid (not shown). These misclassified particles universally lack Poisson spots. Thus, large particles within that temperature range are indeed ice-dominated but to a lesser extent than what the PSDs suggest. We also examined 200 randomly sampled 2D-S images of particles with $D_{eq} > 1$ mm from temperatures between 5°C and 40°C and found that all particles are classified as ice but are actually liquid (not shown). These particles have an elongated shape due to being distorted by the instrument inlet, lack a Poisson spot, and often touch the edge of the image buffer so that they are partially cut-off. We

note that classification skill decreases in general for particles approaching 1 mm because particles of this size are less likely to be fully imaged by the instrument. Thus, we caution that phase classifications for the largest particles may be less certain and that the 5°C to 40°C temperature range is particularly affected by misclassifications of large drops as ice. We note that there





are so few of these large misclassified liquid particles at warm temperatures that they do not show up in Figure8c, which uses a threshold of 100 particles for each 2D histogram bin.

## 5.2 Cloud phase heterogeneity

We also use UWILD's classifications and confidences to compute a 1-Hz estimate of sub 1-Hz cloud phase heterogeneity. Cloud phase heterogeneity, or the degree to which liquid and ice particles are evenly mixed within mixed-phase clouds, can influence cloud radiative (Sun and Shine, 1994) and thermodynamic properties (Korolev et al., 2017). It may also modulate the rates of certain mixed-phase processes such as the Wegener–Bergeron–Findeisen process (Tan and Storelvmo, 2016), and has implications for how those processes should be parameterized in microphysics models. Most studies of cloud phase heterogeneity from in-situ observations have focused on 1-Hz data (Korolev et al., 2003; D'Alessandro et al., 2019; Field et al., 2004; Cober et al., 2001a). Field et al. (2004) used PSDs from a Small Ice Detector in combination with other in-situ measurements to identify cloud phase and found that segments as short as 100-m could contain both liquid and ice. They used 1-Hz data but could investigate relatively small length scales due to low aircraft speeds (100-120 m s$^{-1}$). Here, we use single particle phase classifications to investigate sub 1-Hz cloud phase heterogeneity and we identify mixed-phase periods on the meter-scale.

From the single particle classification data, we derive an estimate of sub 1-Hz heterogeneity by considering whether adjacent particles in the 2D-S image buffer are of the same phase or of different phases, which we denote a phase 'flip' (from I-L or from L-I). If, for a 1-second period, there are many phase flips given the number of particles, that sample is more heterogeneous than one where there are few (or no) phase flips for a population of particles. We leverage the fact that our classifications are probabilistic in determining phase flips and create a probabilistic phase flip prediction as well. Given two adjacent particles $p_1$ and $p_2$:

$$P(\text{flip}) = P(p_1 = \text{ice}) \times P(p_2 = \text{liquid}) + P(p_1 = \text{liquid}) \times P(p_2 = \text{ice})$$

We estimate the most likely number of flips over all particles within a given sample by adding these probabilities together. Thus a hypothetical sample containing 100 particles may have between 0 flips (completely homogeneous sample with 100% classification confidence on all particles) and 99 flips (particles are alternating 100% likely ice and 100% likely liquid), although both of these extremes are unlikely with our probabilistic estimate. A limitation of this heterogeneity estimate is that it implicitly assumes that phase flip probabilities are independent. An advantage of this metric is that it avoids using particle mass to compute phase heterogeneity. Ice particle mass estimated from 2D-S images can vary over an order of magnitude depending on the assumed mass-dimensional relationship (Wu et al., 2019) and more reliable measurements from a Nevzorov instrument with a deep cone (Korolev et al., 2003) were not available during SOCRATES.

We create a 1-Hz heterogeneity measure, which we refer to as the phase flip fraction, by dividing the number of probabilistic phase flips described above by the total number of particles imaged by the 2D-S within one second. We implicitly assume that all unclassified particles, which are mainly particles with fewer than 25 pixels and D$_{\text{eq}} < 0.1$ mm, are liquid, which is a reasonable extrapolation from our PSDs (Figure 10f) as only 1% of the smallest particles are classified as ice. However, this





assumption may lead to an underestimate in phase flips for the coldest samples ($-40°$C$<$ T $<-20°$C) where there are similar
numbers of small droplets and small ice crystals.

Figure 11 shows a 2D histogram with phase flips on the x-axis and the total number of 2D-S imaged particles on the y-axis,
with white lines indicating lines of constant phase flip fraction. We see two distinct modes in this phase space. 1-second samples
with total particle counts below 1000 typically have between 0.02 and 0.5 flips per particle, whereas 1-second samples with
total particle counts between 3000 and 30,000 typically have between 0.002 and 0.0005 flips per particle. In the SOCRATES
dataset, high total particle counts generally indicate the presence of many small droplets. Since these samples are dominated
in number by the liquid phase they have low phase heterogeneity.

Figure 12 shows two examples of meter-scale phase heterogeneity. The statistics of the 1-second periods containing the
plotted segments are included to the right of the image strips. Grey lines bound each particle and all particles that are large
enough to be classified and are not suspected artifacts are labeled with their UWILD classifications and the model confidence
in parentheses. Red labels are used for ice classifications with confidence $\geq 75\%$, blue labels for liquid classifications with
confidence $\geq 75\%$, and purple labels for ice or liquid classifications with confidence $< 75\%$.

In the example in the first row, there is a pocket of small droplets surrounded by large ice crystals within a 3.4-m segment of
cloud. There are three particles within the pocket that have low model confidence and thus may be small droplets or small ice
crystals. This example resembles the conditionally mixed-phase condition described by Korolev et al. (2017), where the cloud
is single phase if you look at a small enough length scale. For the segment of cloud shown, the length scale is approximately 1
meter. The second row shows alternating liquid and ice particles of similar size within a 2.8-m segment of cloud. Here, there is a
larger proportion of low confidence particles that could be either liquid or ice. This is a result of UWILD's tendency, described
in Section 4, to classify some large liquid drops as ice. There is a sequence of four particles classified as I-L-L-I, all with high
confidence, towards the center of the image strip. This example more closely resembles the genuinely mixed-phase condition
described by Korolev et al. (2017) because the phase changes every one or two particles. The difference in the heterogeneity
between the two regions is captured in the 1-second flips per particle, which is about twice as high for the period in the bottom
row than it is for the period in the top row. However, there may be fewer or greater flips per particle in the periods shown than
in the 1-second periods if the periods shown are not representative of the entire 1-second periods.

## 6   Conclusions

In-situ observations of Southern Ocean cloud phase, vital for evaluating simulations and remote sensing products, were sparse
prior to the Southern Ocean Clouds, Radiation and Aerosol Transport Experimental Study (SOCRATES) campaign in January-
February of 2018. The SOCRATES dataset includes nearly 6 million 2D-S shadow images of particles with 25 or more pixels
and with area equivalent diameters ($D_{eq}$) greater than 0.1 mm, which are good candidates for single particle phase classification.
Here, we introduce the University of Washington Ice-Liquid Discriminator (UWILD), a phase classification algorithm that
takes a random forest approach, and show that it outperforms two existing phase classification algorithms which have been
applied to 2D-S images from SOCRATES. In particular, UWILD has the flexibility to identify both liquid-dominated and





ice-dominated regions in the dataset, whereas the other two algorithms both demonstrate strong biases in favor of one phase. UWILD also returns a model confidence for each classification, which is invaluable for computing uncertainties on variables derived from its classifications and for filtering out particles that have a higher likelihood of being misclassified. We believe that

the performance of UWILD is limited largely by the SOCRATES 2D-S data quality (many particle images are out-of-focus) and the challenges of building a train-test-validation (TTV) set, and not the choice of machine learning model. Thus, we would expect to see only modest gains in performance from employing more sophisticated machine learning models.

Since many hydrometeors within mixed-phase clouds can be unidentifiable by eye, we use in-situ observations of atmospheric parameters to select liquid-only and ice-dominated periods within SOCRATES to build a TTV set. If we had limited

the TTV set to particles identifiable by eye, as most studies applying machine learning to airborne probe images have done, the TTV set would not be representative of many particles observed during SOCRATES. There are a small number ($\sim$ 0.3%) of liquid particles in the ice-dominated training data, which may contribute to the misclassification of some large liquid drops as ice. This bias, which is the most prominent one demonstrated by UWILD, is shared by Holroyd and by the automated PHIPS algorithm which uses scattering phase functions to classify phase. Thus, we posit that identifying large liquid drops is a major

outstanding problem in hydrometeor phase discrimination within mixed-phase clouds. We also note that large liquid drops that are misclassified by UWILD typically have model confidence below 75% or are touching the edge of the image buffer, and can be easily filtered out of the dataset if desired.

We use classifications and confidence from UWILD to generate particle size distributions (PSDs) for the whole SOCRATES dataset and find that hydrometeors with $D_{eq}$ greater than 0.15 mm are ice-dominated and that smaller particles are liquid-

dominated, which is in agreement with phase classifications from the PHIPS. We also develop a novel estimate of sub 1-Hz phase heterogeneity by tallying the number of probabilistic phase flips per particle within 1-second periods, which we refer to as the phase flip fraction. This particle number-based approach to estimating heterogeneity avoids uncertainties in estimating particle mass. We use the phase flip fraction to identify two periods in the SOCRATES dataset exhibiting meter-scale phase heterogeneity.

SOCRATES sampled mixed-phase Southern Ocean clouds with the goal of improving their representation in climate models. Here, we process SOCRATES in-situ observations from the 2D-S to create datasets that can be more effectively compared with atmospheric models and remote sensing datasets. Liquid and ice PSDs can be used to evaluate microphysical model output. Single particle phase classifications can be coarsened and compared with cloud top phase products from Himawari and MODIS. Single particle classifications from UWILD, and the 1-Hz variables that we derive from them, are also useful for informing

the development of microphysics parameterizations. Large droplets are necessary for Hallett-Mossop rime splintering and droplet freezing and estimating their concentrations from liquid classifications of 2D-S images can aid in understanding the ice-production process. Additionally, mixed-phase processes such as the Bergeron-Findeisen mechanism for rapid ice growth and Hallett-Mossop rime splintering may operate more slowly in conditionally mixed-phase clouds, which have a small number of flips per particle, than genuinely mixed-phase clouds, which have a large number of flips per particle. UWILD is publicly

available and we encourage readers to adapt it for other in-situ datasets to examine processes controlling phase partitioning in mixed-phase clouds globally.



*Code and data availability.*

The 2D-S (https://data.eol.ucar.edu/dataset/552.009), VSCEL (Diao, 2020), PHIPS (Schnaiter, 2018a, b) and aircraft (EOL, 2019) data used in this study are found on the NCAR EOL data archive. The software packages used to process the OAP data (McFarquhar et al., 2018) and to run the random forest model (Mohrmann et al., 2021) are publicly available as GitHub repositories. The data containing 1-Hz phase-partitioned PSDs and phase flip fraction estimates will be made publicly available on the NCAR EOL data archive upon submission.

**Appendix A**

**A1**

We show histograms of 14 particle parameters for the TTV set and the 14 SOCRATES flights analyzed in this study, in Figure A1. We do not include the parameter touching_edge here because it is binary. The histograms are plotted on a log scale so that the tails of their distributions are visible. About 0.6% of particle images from the SOCRATES dataset that we analyzed here are out of sample. This means that at least one particle parameter has a value that is outside of the range of the TTV set. Such particles are usually out of sample because they are larger than all of the particles in the TTV set. We examined 45 randomly sampled images from the SOCRATES dataset that have an area-equivalent diameter ($D_{eq}$) greater than all particles in the TTV set. These particles are mainly heavily rimed aggregates. All of them are clearly frozen and UWILD classifies them as such. Thus, we do not believe that the small percentage of out of sample particles present in the SOCRATES dataset reduces the performance of UWILD.

Area ratio is greater than 1.0 about 25% of the time This is because a correction is applied to the calculation of the maximum dimension following Korolev (2007) when the diode at the center of the minimum circle enclosing the particle is unshadowed, which is the case for particles featuring Poisson spots. In these cases, the area of a circle with a diameter equal to the corrected maximum dimension can be smaller then the projected area of the particle.

*Author contributions.* RA prepared the manuscript with contributions from all co-authors, and lead the analysis of the phase classifications from the three different algorithms for the whole SOCRATES dataset (Section 4). JM lead the development of UWILD, with contributions from JL and IH, and computed size distributions and phase heterogeneity metrics from UWILD's classifications and model confidences (Section 5). JF processed the 2D-S data to obtain particle features used in the machine learning model and contributed expertise on the 2D-S instrument and the SOCRATES campaign. JL and IH trained and tuned UWILD and evaluated its performance on the test set (Section 4). In particular, JL computed F1 scores for different particle categories and IH computed permutation feature importance. RW provided continuous feedback and guidance during the study. MD re-calibrated the water vapor data from the VCSEL instrument and provided comments for the manuscript.



*Competing interests.* No competing interests are present.

*Acknowledgements.* The authors acknowledge all SOCRATES scientists for collecting the data used in this study. The authors are grateful to Emma Järvinen and Fritz Waitz for their help with interpreting the PHIPS data and Wei Wu for his work on the initial 2D-S single particle phase classification. UW participants acknowedge funding from NSF Atmospheric & Geospace Sciences #1660609. MD acknowledges the

560   support from National Center for Atmospheric Research (NCAR) Advanced Study Program (ASP) Faculty Fellowship in 2018 and NSF Office of Polar Programs grant #1744965.



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





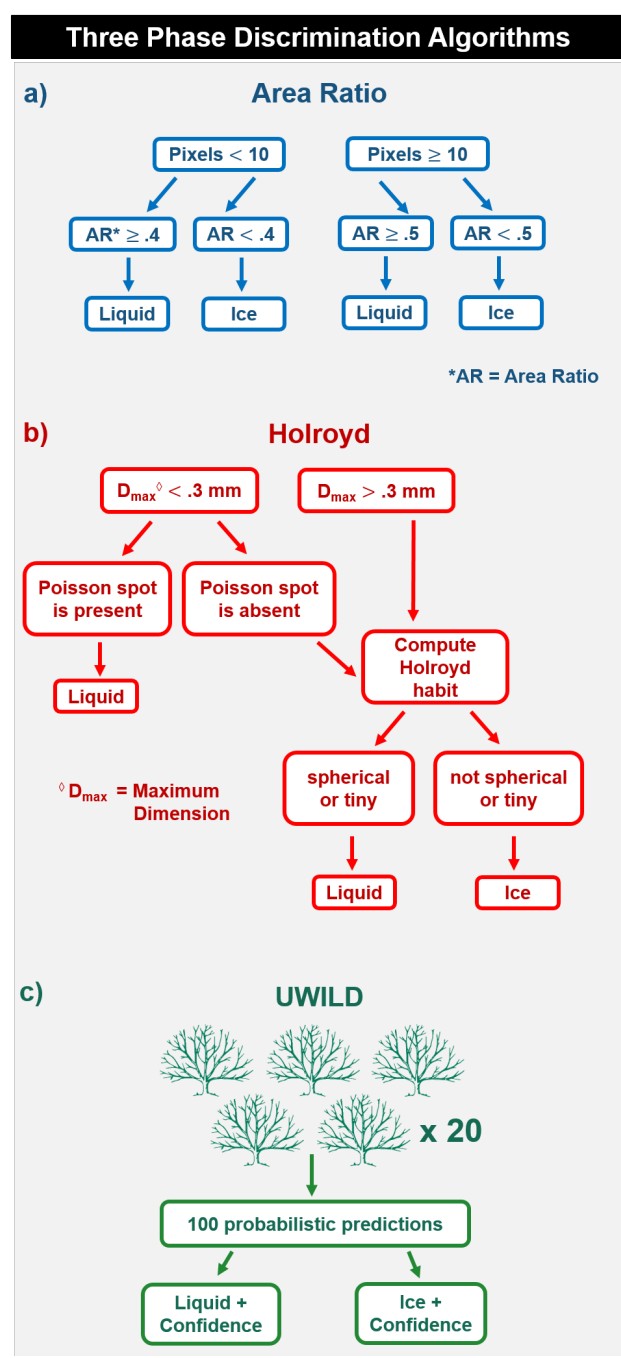

**Figure 1.** Schematic comparing the three phase classification algorithms compared in this study. Using single particle properties derived from the 2D-S probe as input, each panel describes the algorithm decision tree used to classify a particle as liquid or ice.



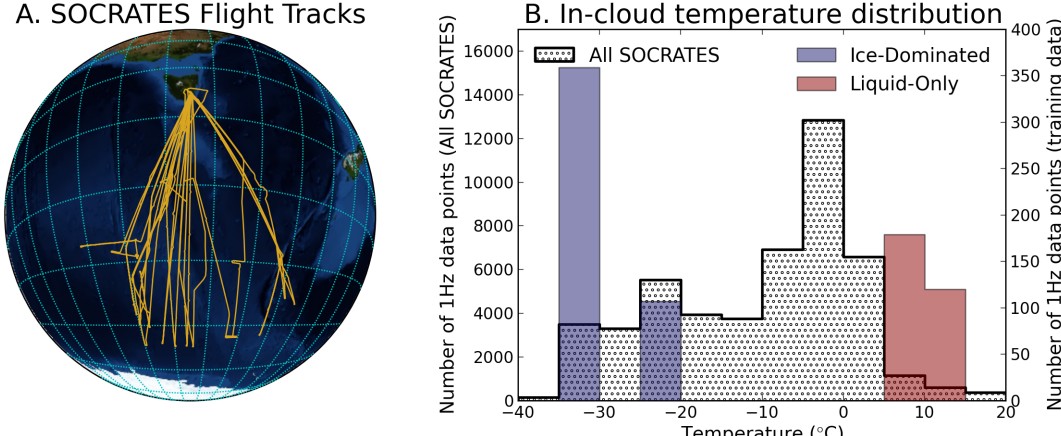

**Figure 2.** A. SOCRATES flight tracks from RF01-RF14 (the flights analyzed in this study). B. 1-Hz in-cloud temperature histograms for RF01-RF14 (hatched, left y-axis), ice-dominated data from the TTV set (blue, right y-axis) and liquid-only data from the TTV set (red, right y-axis). In-cloud data includes all samples with at least 1 particle that satisfy our selection criteria (described in Section 2).

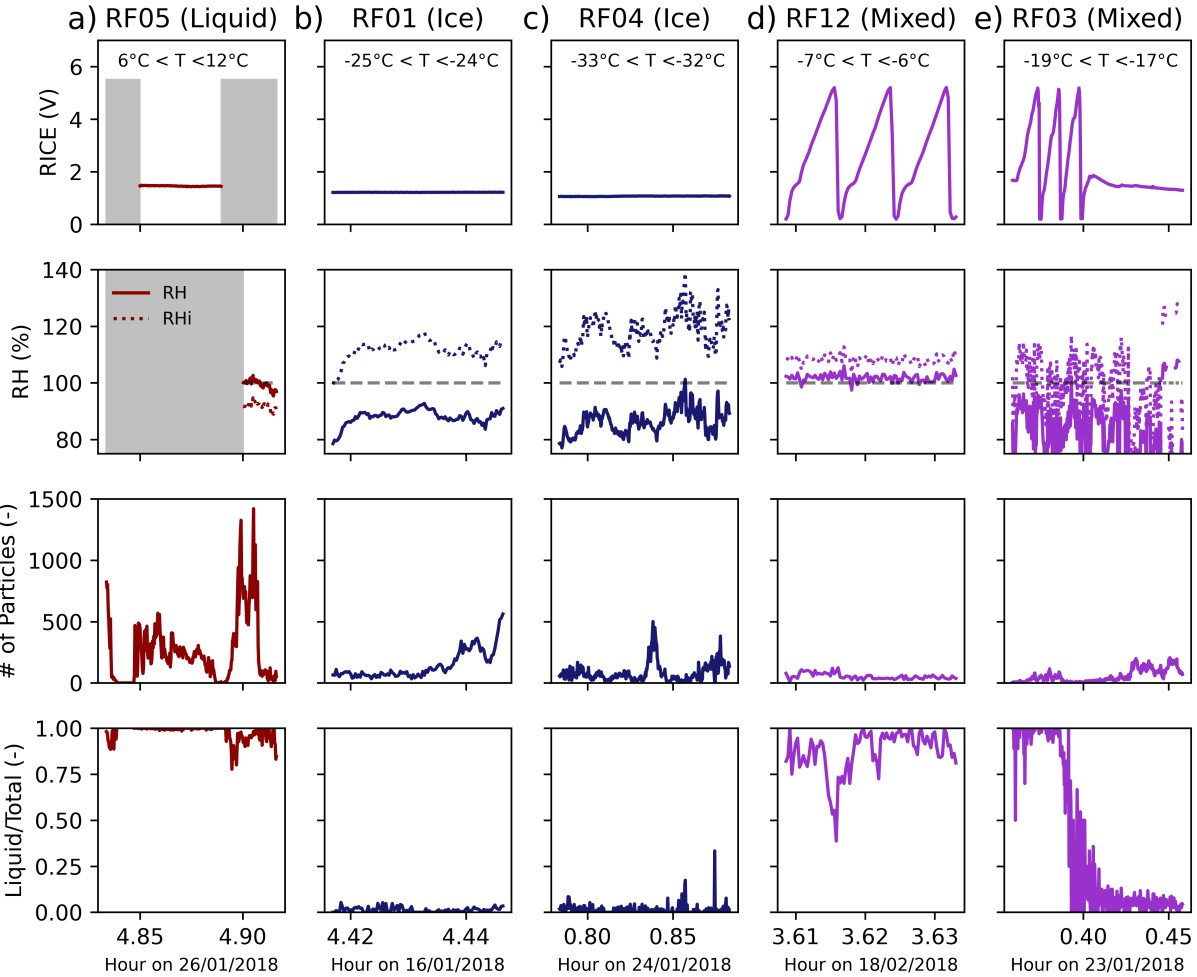

**Figure 3.** Time series of atmospheric parameters and particle properties for the liquid-only period (a) and the two ice-dominated periods (b,c), and the two example mixed-phase periods (d,e). Voltage from the RICE (row 1), relative humidity with respect to liquid (solid line) and ice (dashed line) (row 2), the number of particles that satisfy our selection criteria (row 3), and liquid fraction as determined by the UWILD algorithm (row 4) are shown for each flight period. A RICE response is expected when the supercooled liquid water content exceeds 0.01 g m$^{-3}$. Missing data are indicated with grey. The temperature range sampled in each period is shown on the plot of RICE voltage in the top row.



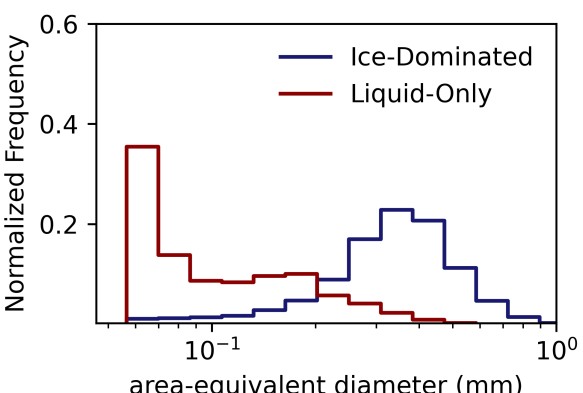

**Figure 4.** Distributions of particle size for the liquid-only and ice-dominated data from the TTV set.

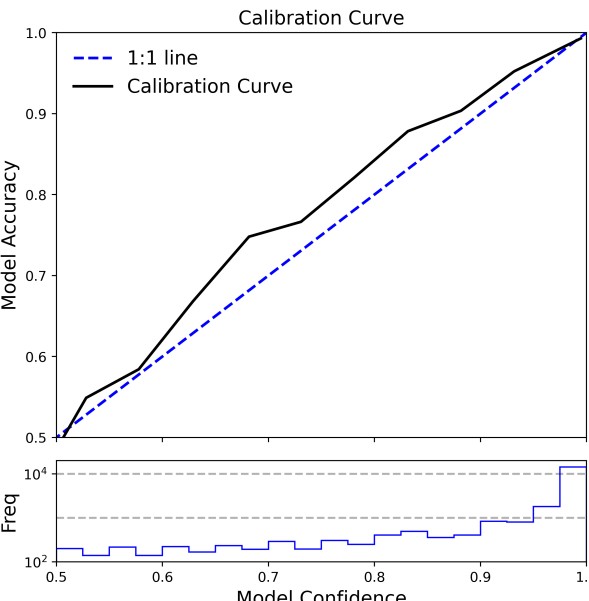

**Figure 5.** UWILD confidence is plotted against UWILD accuracy with a black solid line (this is referred to as the calibration curve) in the top row and a histogram of model confidence is shown in the bottom row on a shared x-axis, for the test set. The proximity between the calibration curve and the one-to-one blue dashed line implies that UWILD confidence can be used as a proxy for the uncertainty in UWILD's classifications.



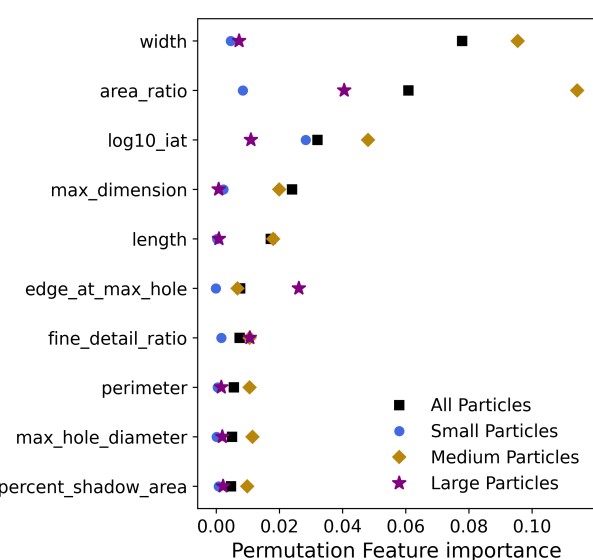

**Figure 6.** Permutation feature importance for the ten most important features is shown for the three size classes. Features are shuffled ten times and the mean feature importance from the ten trials is shown here.

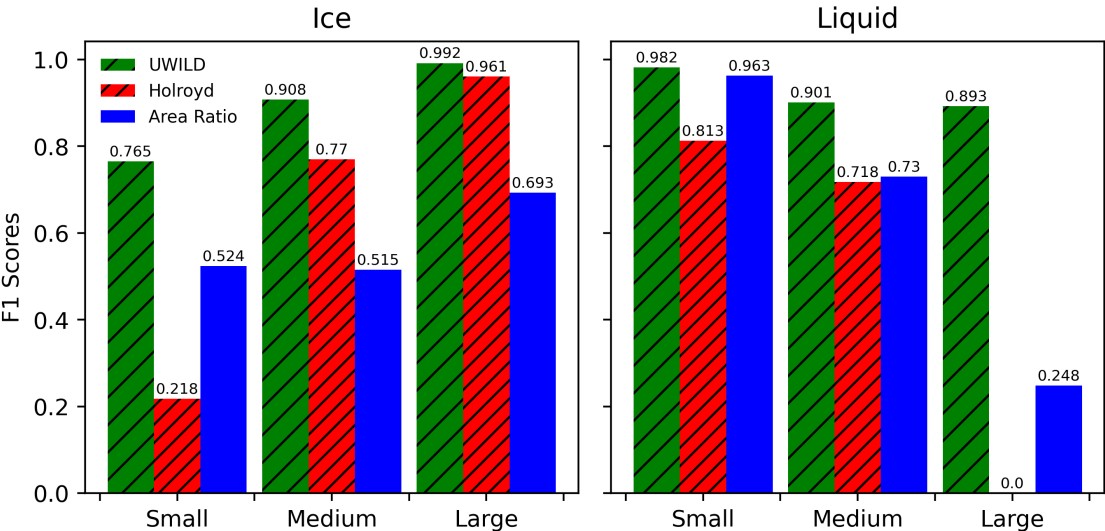

**Figure 7.** F1 scores are shown for UWILD (green), Holroyd (red) and Area Ratio (blue) for different size classes. F1 scores for small particles may be underestimated due to the presence of liquid drops in the primarily ice training/validation/test set.

**Figure 8.** 2D histograms of the number of particles meeting our criteria (row 1), UWILD confidence (row 2), and phase classifications for the three algorithms (rows 3-5) are shown in temperature-particle size phase space in the right column and relative humidity-particle size phase space in the left column. A threshold of 100 total particles per 2D histogram bin is used for all plots. SOCRATES flights RF01-RF14 are included.





**Figure 9.** Randomly sampled images are shown for the five regions overlaid on the temperature-particle size phase space in Figure 7. UWILD confidence is displayed above each particle and UWILD phase (green), Area Ratio phase (blue), and Holroyd phase (red) are shown to the right of each particle. The time dimension is vertical and the photodiode dimension is horizontal.



**Figure 10.** Particle size distributions are averaged over 1-Hz data from SOCRATES flights RF01-RF14 and shown for six different temperature ranges. Dashed lines are deterministic predictions from UWILD. Solid lines and shaded areas are medians and interquartile ranges, respectively, for 30 bootstrapped samples generated from UWILD confidences. Red dashed lines separate the three different size classes.

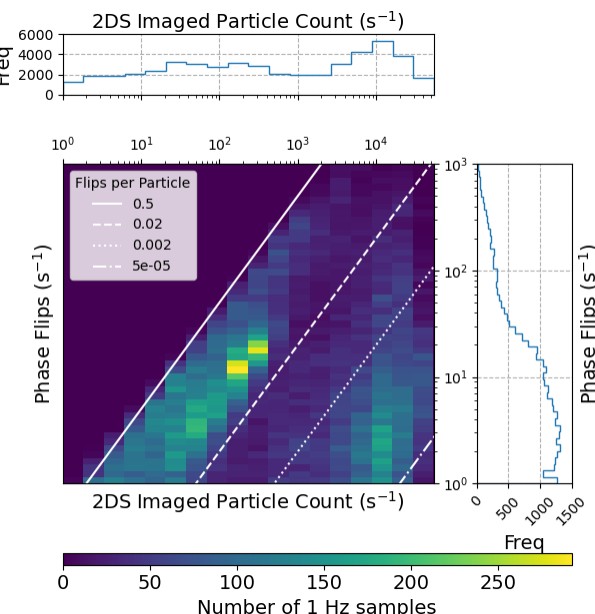

**Figure 11.** A 2D histogram of 1-Hz particle phase flips (y-axis) and 1-Hz 2D-S imaged particles (including both successfully classified particles and unclassifiable particles) is shown in the top row. A 1D histogram of 1-Hz 2D-S imaged particles is shown on top, on a shared x-axis, and a 1D histogram of 1-Hz phase flips per second is shown on the right, on a shared y-axis. A value closer to the upper left of the plot indicates a higher degree of particle heterogeneity. White lines indicate lines of constant heterogeneity for varying particle counts. Data are from all SOCRATES flights RF01-RF14.

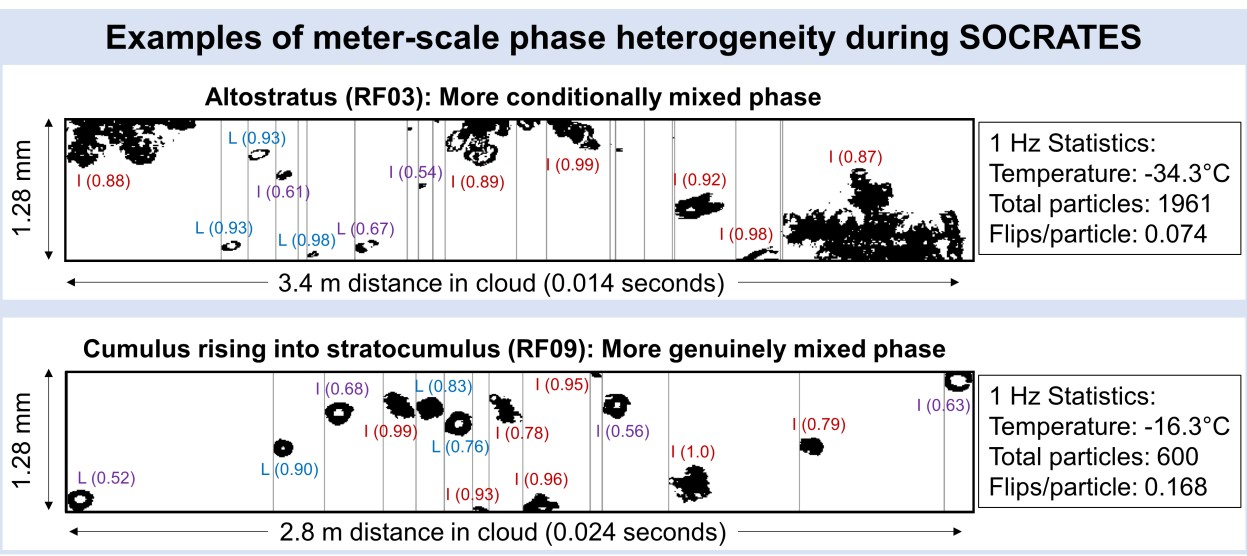

**Figure 12.** Short periods from two different cloud types highlight instances of meter-scale heterogeneity in the SOCRATES dataset. The photodiode dimension is vertical and the time dimension is horizontal, which is the reverse of Figure 9. Particle labels show UWILD classifications and UWILD confidences in parentheses. Red indicates ice classifications with confidences greater than 75%, blue indicates liquid classifications with confidences greater than 75% and purple indicates ice or liquid classifications with confidences less than 75%





**Figure A1.** Histograms of particle features are shown for the TTV set (black) and for SOCRATES flights RF01-RF14 which are analyzed in Sections 4 and 5. touching_edge is not shown here because it is a binary variable.



**Table 1.** Description of each particle feature used in the random forest model

| Particle feature | Description [units] |
|---|---|
| length | Pixels in time dimension |
| width | Pixels in photodiode dimension |
| particle area | Projected area from the number of shadowed pixels (1 pixel = 0.0001 mm$^2$) [mm$^2$] |
| perimeter | Perimeter following the particle boundary [mm] |
| max_dimension | Maximum dimension of the smallest circle enclosing the particle [mm] |
| eq_diameter | Area equivalent diameter: diameter of a circle with an area equal to the projected area [mm] |
| area_ratio | Ratio between the projected area and that of the smallest circle enclosing the particle [#] |
| percent_shadow_area | Ratio between the projected area and the length×width [#] |
| touching_edge | Binary depicting whether the particle is entirely within the photodiode array [0] or touches an edge of the array [1] |
| max_top_edge_touching | Maximum number of times the top diode is shadowed in succession [#] |
| max_bottom_edge_touching | Maximum number of times the bottom diode is shadowed in succession [#] |
| edge_at_max_hole | Number of pixels between edges of the particle for the slice containing the largest gap inside the particle [#] |
| max_hole_diameter | Diameter of the largest hole inside the particle [#] |
| fine_detail_ratio | Ratio between the perimeter*maximum dimension and projected area following Holroyd (1987) [#] |
| log10_iat | Log of the inter-arrival time between particles [$\log_{10}$(s)] |





**Table 2.** Number of particles by size class in the training, validation and test sets

| Phase | Particle Size Class | Training | Validation | Test |
|---|---|---|---|---|
| Ice | Small (0.056 mm $\leq D_{eq} <$ 0.1 mm) | 1.7k | 0.59k | 0.6k |
| | Medium (0.1 mm $\leq D_{eq} <$ 0.3 mm) | 12.6k | 4.2k | 4.2k |
| | Large ($D_{eq} \geq$ 0.3 mm) | 18.6k | 6.2k | 6.2k |
| Liquid | Small (0.056 mm $\leq D_{eq} <$ 0.1 mm) | 20.1k | 6.7k | 6.7k |
| | Medium (0.1 mm $\leq D_{eq} <$ 0.3 mm) | 11.4k | 3.8k | 3.8k |
| | Large ($D_{eq} \geq$ 0.3 mm) | 1.5k | 0.5k | 0.5k |





**Table 3.** Total particles and number of particles classified as liquid from the three phase discrimination algorithms for each size class. Numbers in parentheses are the percentage of total particles classified as liquid. SOCRATES flight RF01-RF14 are used.

| Particle Size Class | Total | UWILD | Holroyd | Area Ratio |
|---|---|---|---|---|
| Small (0.056 mm $\leq D_{eq} < 0.1$ mm) | 2788219 | 2641062 (94.7) | 1867682 (70.0) | 2563437 (91.9) |
| Medium (0.1 mm $\leq D_{eq} < 0.3$ mm) | 1710833 | 674648 (39.4) | 636655 (37.2) | 1252175 (73.2) |
| Large ($D_{eq} \geq 0.3$ mm) | 1265670 | 34040 (2.7) | 0 (0.0) | 614144 (48.5) |
| All particles | 5764722 | 3349750 (58.1) | 2504337 (43.4) | 4429756 (76.8) |