# Peer review of "The University of Washington Ice-Liquid Discriminator (UWILD) improves single particle phase classifications of hydrometeors within Southern Ocean clouds using machine learning"

_Atmospheric Measurement Techniques, 2021_

## Author Comment (AC1)

We thank both of our reviewers for their insightful and thorough feedback. We particular thank reviewer 1 for encouraging us to explore how inter-arrival time is being used for phase classification and reviewer 2 for suggesting that we create a hand-labelled test set to evaluate the identification of ice crystals at warmer temperatures. We have responded to all comments from both reviewers below. All line numbers refer to the edited manuscript (not the track changes version).

**Response to reviewer 1**

1) The primary reason that we compare UWILD with the Area Ratio and Holroyd algorithms is that these two methods were used to process SOCRATES data. We wanted to investigate how our product compares with the two products currently available in the SOCRATES data archive, so that users can make informed decisions about which dataset to use for their work. Hunter et al. 1983 classifies ice crystals into different habits and does not have a liquid or spherical category, so we wouldn't be able to use their algorithm out of the box for phase classification. Rahman et al. 1981 does include a raindrop category and is thus suitable for comparing with UWILD. We have not been able to obtain a copy of Duroure 1982 either in English or the original French, so we cannot assess whether or not it is appropriate for phase classification.

2) We limited our features to particle properties already computed by the University of Illinois and Oklahoma post-processing software, which are listed here: https://github.com/joefinlon/UIOPS/wiki/Output-Data. We added a sentence to the manuscript explaining this (lines 134-135). We also added a histogram of the common logarithm of inter-arrival time (log10(iat)) for liquid and ice particles from our TTV set to the manuscript in the new Figure 3a, and we added a discussion of that figure and the use of inter-arrival time for phase classification to Section 2.1 (lines 139-148). The ice-dominated particles generally have inter-arrival times between  $10^{-4}$  and  $10^{-2}$  seconds, whereas the liquid particles generally have inter-arrival times between  $10^{-6}$  and  $10^{-3}$  seconds. This difference in inter-arrival times between liquid and ice particles is what motivates us to use inter-arrival time for phase discrimination. The relatively long inter-arrival times for ice crystals indicates that the ice crystals in the TTV set are not dominated particles ( $10^{-7}$  and  $10^{-5}$  seconds) is likely populated by shattered artefacts but it represents just 7% of the ice-dominated particles. Thus, we expect that shattered artefacts only minimally impact UWILD's use of inter-arrival time for phase discrimination.

Another motivation for including particle inter-arrival time is that as you go towards smaller sizes, there is less information stored within the particle images and it is more difficult to resolve morphological features. Thus, it becomes increasingly desirable to use a variable that is not derived from the particle images in addition to the image-derived variables. Figure 6 hows that  $log_{10}(iat)$  is the most important feature for the small particles. Moreover, it is more than twice as important as any other feature. To further emphasize the importance of particle inter-arrival time to the phase classification of small particles, we re-trained UWILD without inter-arrival time. Figure 1 here compares the feature importance between our final version of UWILD (left) and a version of UWILD that does not use  $log_{10}(iat)$  is removed but is dramatically different for the small and medium particles. For these, several features become much more important, with the largest changes seen in the fine\_detail\_ratio.

Figure 1: Permutation feature importance for the ten most important features is shown for the three size classes. Features are shuffled ten times and the mean feature importance from the ten trials is shown here. The plot on the left is the same as Figure 6 in the manuscript. The plot on the right shows the results from a version of UWILD that does not use  $log_{10}(iat)$  as a feature.

Figure 2 here compares the F1 scores for our final version of UWILD (top) and a version of UWILD that does not use  $log_{10}(iat)$  as a feature (bottom). Note that only the green bars, which show the results from UWILD, change between the two plots. The F1 scores for small ice particles plummet from 0.765 to 0.475 when  $log_{10}(iat)$  is removed. The F1 scores for all other phase and size classes decrease only slightly.

We added a description of this sensitivity test to lines 146-148 and 307-309 of the manuscript.

3) The use of synthetic data has been fruitful in the past, especially for studies focusing on the relationship between a specific feature (e.g., area ratio) and a classification; those studies benefit from a well-controlled feature space. A primary goal for our study was to build a classifier that works on real-world particles, including those with ambiguous features influenced by various environmental properties and microphysical processes (e.g., riming). Further, it is not obvious how to create a synthetic dataset that explicitly resolves particular features that do not have an obvious relationship to phase a priori (e.g., inter-arrival time).

Another reason to use a synthetic dataset is in the absence of a reliable real-world dataset. Here we have two good real-world datasets, including the TTV set used in the bulk of the manuscript (generated based on aircraft measurements from particular time periods) as well as the manually-labeled dataset generated in response to reviewer 2's first comment.

4) We have added Figure A1 to Appendix A to visualize our particle parameters using two examples of real particles from the SOCRATES dataset.

**Response to reviewer 2**

1) We agree that there are differences in the average ice crystal properties between the TTV set and the rest of the SOCRATES data. Figure 3 here shows 100 randomly sampled particles from the ice TTV set (top) and 100 randomly sampled particles occurring at temperatures warmer than -23°C (bottom). In both cases, particles with area equivalent diameters between 0.2 and 0.8 mm are included. There are more large bullet rosettes in the ice TTV set and there are more small columns amongst the warmer ice particles. The ice particles in both subsets are dominated by quasi-spherical and irregular ice crystals. Although the ice particles included in the ice TTV set were sampled at temperatures between -24°C and -33°C, they occur within altostratus clouds that generate supercooled liquid particles at the cloud top, as discussed in Section 4 of the manuscript. For this reason, rimed ice crystals are observed even at very cold temperatures. Although the particles look more similar across the range of temperatures sampled during SOCRATES than one might expect, we nevertheless agree with the reviewer that additional work is needed to evaluate UWILD's ability to identify ice particles in warmer clouds, which were the primary target of the SOCRATES campaign.

We have taken the reviewer's suggestion and generated a hand-labelled dataset for additional comparison between the three algorithms. We randomly sampled 1000 particles at temperatures warmer than -23°C (warmer than any data in the TTV set) with area equivalent diameters between 0.2 and 0.8 mm. 100 of those particles are shown on the bottom in Figure 3 here. We chose this size range because smaller particles are difficult to classify by eye. We also remove the largest particles because it is difficult to show a large number of them in one file and because we know that UWILD will always classify them as ice. Four of the co-authors made manual classifications of the particles prior to looking at any of the algorithms' classifications and without discussing the particles with each other.

Particles labeled the same by all four labelers were considered either ice or liquid, while particles on which there was any disagreement between labelers were discarded, leaving a test set of 939 particles (861 ice particles and 78 liquid particles) of unambiguous "ground truth". This set was used to additionally evaluate performance of the three algorithms. F1 scores are as following for the full dataset: UWILD 0.63, Holroyd 0.44, Area Ratio 0.26. This is broken down by ice and liquid in Figure 4 here. Overall, F1 scores are lower than for the original test data, which is expected; these size and temperature ranges are the ones for which there is the most ambiguity, and lowest confidence for UWILD (see manuscript Figure 8b). However, UWILD still outperforms both existing classifications. Figure 5 here shows the confusion matrices for each classification, highlighting the types of error of each scheme: Area Ratio labels much ice as liquid, Holroyd labels much liquid as ice, and UWILD does a little of both. In response to the reviewer's main concern, UWILD has little issue correctly identifying the bulk of the ice particles as ice (as shown by the high ice F1 scores in Figure 4 here), though the strong class imbalance in the test set (with 11 times as many ice particles) means that the small number of misclassifications results in a lowered performance in correctly identifying liquid particles.

We added a summary of this effort to lines 310-320 of the manuscript.

2) We changed "optical resolution" to "pixel resolution" to be more consistent with Lawson et al. (2006) and the rest of our manuscript.

3) We have added an explanation about how the RICE instrument detects supercooled water to lines 161-166 in the manuscript.

4) The reviewer is correct, and it is possible to add an additional variable to the final layer of a CNN-type classification, and this isn't the strongest reason to avoid using a deep learning model. The issue then resides in evaluating the relative importance of this variable to other features of the input particle, as the architecture of the model does not treat calculated input variables equally (unlike the random forest approach). This point is better addressed in the sentence following, so we have removed the one the reviewer took issue with.

5) We clarified this point in lines 341-343 and lines 359-360 of the manuscript. We expect that if the liquid fraction is not already 1.0 at a temperature of 0°C then it will increase as temperature increases above 0°C, and that it will be equal to 1.0 at temperatures > 5°C. Both UWILD and Holroyd deviate from this expected behavior.

6) We accidentally used an old version of Figure 10 in the submitted manuscript. In the new version, letters have been added and the dashed and solid lines have been switched. Thank you for catching this error!

7) We agree that the term "ice-production process" is too vague. We changed "can aid in understanding the ice-production process" to "is useful for determining an upper bound for the rate at which they can produce new ice particles".